# Metformin: From Diabetes to Cancer—Unveiling Molecular Mechanisms and Therapeutic Strategies

**DOI:** 10.3390/biology13050302

**Published:** 2024-04-27

**Authors:** Emilia Amengual-Cladera, Pere Miquel Morla-Barcelo, Andrea Morán-Costoya, Jorge Sastre-Serra, Daniel Gabriel Pons, Adamo Valle, Pilar Roca, Mercedes Nadal-Serrano

**Affiliations:** 1Grupo Metabolismo Energético y Nutrición, Instituto Universitario de Investigación en Ciencias de la Salud (IUNICS), Universitat de les Illes Balears, Ctra. de Valldemossa, km 7.5, 07122 Palma, Illes Balears, Spain; emilia.amengual@uib.es (E.A.-C.); andrea.moran@uib.es (A.M.-C.); adamo.valle@uib.es (A.V.); 2Instituto de Investigación Sanitaria Illes Balears (IdISBa), 07120 Palma, Illes Balears, Spain; pere.morla@uib.es (P.M.M.-B.); jorge.sastre@uib.es (J.S.-S.); d.pons@uib.es (D.G.P.); mercedes.nadal@uib.es (M.N.-S.); 3Grupo Multidisciplinar de Oncología Traslacional, Instituto Universitario de Investigación en Ciencias de la Salud (IUNICS), Universitat de les Illes Balears, Ctra. de Valldemossa, km 7.5, 07122 Palma, Illes Balears, Spain; 4Centro de Investigación Biomédica en Red Fisiopatología de la Obesidad y Nutrición (CIBERobn, CB06/03/0043), Instituto de Salud Carlos III, 28029 Madrid, Spain

**Keywords:** metformin, diabetes, mitochondria, breast cancer, colorectal cancer

## Abstract

**Simple Summary:**

Metformin, a common anti-diabetic drug, is being explored for its potential in managing cancer, especially in breast and colorectal cancer. It works by affecting mitochondrial respiration, causing energy deficits and molecular changes that help control hyperglycemia in type 2 diabetes. Clinical evidence suggests that metformin may prevent cancer in diabetic patients, improving survival outcomes in breast and colorectal cancer. When used in combination with chemotherapy and immunotherapy, metformin shows promising synergistic effects. Ongoing studies explore metformin’s bioavailability, efficacy, and new metformin-based compounds, including those targeting mitochondria, to enhance its anticancer effects.

**Abstract:**

Metformin, a widely used anti-diabetic drug, has garnered attention for its potential in cancer management, particularly in breast and colorectal cancer. It is established that metformin reduces mitochondrial respiration, but its specific molecular targets within mitochondria vary. Proposed mechanisms include inhibiting mitochondrial respiratory chain Complex I and/or Complex IV, and mitochondrial glycerophosphate dehydrogenase, among others. These actions lead to cellular energy deficits, redox state changes, and several molecular changes that reduce hyperglycemia in type 2 diabetic patients. Clinical evidence supports metformin’s role in cancer prevention in type 2 diabetes mellitus patients. Moreover, in these patients with breast and colorectal cancer, metformin consumption leads to an improvement in survival outcomes and prognosis. The synergistic effects of metformin with chemotherapy and immunotherapy highlights its potential as an adjunctive therapy for breast and colorectal cancer. However, nuanced findings underscore the need for further research and stratification by molecular subtype, particularly for breast cancer. This comprehensive review integrates metformin-related findings from epidemiological, clinical, and preclinical studies in breast and colorectal cancer. Here, we discuss current research addressed to define metformin’s bioavailability and efficacy, exploring novel metformin-based compounds and drug delivery systems, including derivatives targeting mitochondria, combination therapies, and novel nanoformulations, showing enhanced anticancer effects.

## 1. Introduction

Metformin is widely recognized as a primary medication in the global management of type 2 diabetes mellitus (T2DM), earning its inclusion on the World Health Organization’s list of essential medicines since 2019 [1]. Its low cost, excellent tolerability, and safety profile, with low risk of hypoglycemia, make it a preferred choice either alone or in combination with other drugs for managing millions of patients with T2DM [2]. Metformin belongs to the biguanide family (1,1-dimethylbiguanida hydrochloride) originating from the plant known as French lilac (*Galega officinalis*) [3]. Historically, this plant has been utilized in Europe since the Middle Ages to alleviate symptoms of diabetes mellitus, owing to its abundance of galegine, an isoprenyl guanidine compound. While monoguanidines and diguanidines exhibit toxicity, biguanides, composed of two N-linked guanidine molecules, have been employed for diabetes treatment since the late 1950s [4]. Metformin gained approval for use in Europe and Canada in 1957, not being introduced in the United States until 1995 [3]. The more potent biguanides, phenformin and buformin, were widely used in the United States and Europe in the 1960s; however, they were withdrawn from the market in the late 1970s due to a higher risk of lactic acidosis compared to metformin (approximately 3–9 cases per 100,000 person-years). Notably, the risk of lactic acidosis is elevated in patients with chronically impaired renal function or acute kidney disease, populations for whom metformin is contraindicated [5]. Consequently, the most prevalent adverse effect associated with metformin is gastrointestinal intolerance [6].

Although metformin has been administered to millions of patients with T2DM for over 60 years, the exact mechanism (or mechanisms) of action remains a subject of debate. Metformin is described as an anti-hyperglycemic agent that does not induce clinical hypoglycemia in patients with T2DM or disturb glucose homeostasis in non-diabetic individuals [2,7]. It is known that metformin primarily acts by suppressing hepatic glucose production, which is increased in individuals with T2DM, through a decrease of 25–40% in the hepatic gluconeogenesis rate [8]. Additionally, some euglycemic-hyperinsulinemic clamp studies suggest it may also have a beneficial effect on insulin sensitivity at the skeletal muscle level, although this effect is not consistently observed across all studies [9,10,11,12]. In recent years, a growing body of evidence points to the gut as a key target of metformin action, promoting glucose utilization, growth/differentiation factor-15 (GDF15) secretion, which reduces appetite, and regulating intestinal microbiota, all of which collectively contribute to its potential benefits (reviewed in (Barroso et al., 2023) [13]).

Controversy surrounds the primary targets of metformin to promote the reduction in hepatic gluconeogenesis. For over 50 years, it has been known that biguanides decrease mitochondrial respiration, thus placing mitochondria at the core of their action [14]. However, the precise molecular targets within this organelle and their subsequent effects are diverse, leading to a plethora of proposed mechanisms, some of which may overlap, in explaining metformin’s antigluconeogenic effects in the liver. Notably, the inhibition of mitochondrial respiratory chain Complex I and mitochondrial glycerophosphate dehydrogenase are prominent among these targets, although other respiratory chain complexes have also been suggested as potential targets [15,16,17]. Metformin’s impact on these targets manifests in a complex array of interconnected effects, including cellular energy deficits, changes in the redox state, the activation of adenosine monophosphate (AMP)-activated protein kinase (AMPK), the inhibition of cyclic adenosine monophosphate (cAMP)-mediated glucagon signaling, allosteric modulation of gluconeogenic enzymes, and epigenetic alterations [18,19]. Untangling which of these events truly drives the reduction in hepatic gluconeogenesis induced by metformin remains a subject of intense debate. Moreover, the fact that many of metformin’s effects seen in in vitro and preclinical studies occur at suprapharmacological concentrations, coupled with discrepancies in effects between acute and chronic administration in in vivo models, further complicates the interpretation of these findings and, consequently, the elucidation of the underlying molecular mechanisms of this drug. Investigation into the underlying molecular mechanisms of action of metformin has garnered significant interest, beyond its applications in diabetes treatment. Metformin’s capacity to alter cellular bioenergetics and modulate crucial aspects of mitochondrial function, such as oxidative stress and apoptosis, has sparked curiosity about its potential repurposing in treating various diseases [18]. Notably, there is growing enthusiasm for exploring metformin’s potential as a therapeutic agent against certain types of cancer, such as breast and colon cancer [20,21,22]. One of the hallmarks of cancer is the reprogramming of cellular energy metabolism, allowing tumor cells to sustain continuous growth and proliferation by substituting the metabolic program typically found in normal tissues. Drugs like metformin, with the ability to exploit specific metabolic vulnerabilities in tumor cells, present a promising avenue for cancer treatment [23]. Indeed, lower incidences of certain types of cancer and/or improved overall survival has been reported in T2DM patients treated with metformin [22,24,25]. Repurposing existing drugs for other diseases offers significant time and cost-saving advantages, as their pharmacokinetics, pharmacodynamics, and safety profiles are already established, thus allowing preclinical studies to be streamlined. Given the considerable interest in metformin, this review aims to delve into the current understanding of its mechanisms of action and its potential applications in breast and colon cancer treatment. Furthermore, we will examine the primary challenges associated with repurposing metformin for cancer therapy and discuss the strategies being contemplated to address these challenges.

### Pharmacokinetics

Metformin is characterized chemically as a highly hydrophilic compound with an acid dissociation constant (pKa) of 11.5, meaning that, at a physiological pH, the drug exists as a monoprotonated cation. The presence of charge at a physiological pH results in the following two main consequences: drug transport across biological membranes involves uptake via specific transporters, and organelles, such as energized mitochondria, can slowly accumulate the drug driven by their transmembrane electrochemical potential (Δψ) [26]. Thus, it is known that the absorption, distribution, and excretion of metformin primarily rely on organic cation transporters (OCTs), multidrug and toxin extruders (MATEs), and plasma membrane monoamine transporter (PMAT) [27,28].

Metformin is typically administered orally and exhibits a low bioavailability, ranging from 40% to 60% [29]. The drug is not metabolized and is excreted unchanged through urine. In the treatment of T2DM, patients are typically prescribed a dosage of 25–30 mg/kg per day, typically divided into two or three oral doses of 500–850 mg each. However, it is important to note that the absorption of metformin is reliant on specific transporters and, therefore, administering higher doses can slow the absorption rate and decrease overall bioavailability [30]. In mouse models, higher doses (200–250 mg/kg) are often necessary due to their more efficient renal clearance, resulting in a shorter half-life of 1–2 h compared to the 4–9 h observed in humans [31]. Biodistribution studies in humans, employing positron emission tomography (PET) with ^11^C-labeled metformin, demonstrate its primary distribution in the small intestine, liver, and kidneys. This distribution pattern aligns with both the expression profile of previously discussed specific transporters and the major target organs (liver and gut), as well as its high rate of renal elimination [27,32]. Following intestinal absorption, metformin attains high concentrations in the portal vein (40–70 µM), leading to the accumulation of higher levels of metformin in the liver than in surrounding organs, as confirmed by PET in in both humans and mice [17,33,34]. It is noteworthy that studies in mice indicate that the liver can reach concentrations even higher than those in portal vein plasma [35].

In this sense, it is important to emphasize that antineoplastic effects of metformin depend on drug concentration within neoplastic tissue. This concentration is influenced not only by plasma level, but also by cellular uptake in cancer cells, which depends on the expression of relevant transporters, including OCT1 [36]. It is worth noting that serum levels of metformin achieved in diabetic patients and in vivo models are in the micromolar range, while in vitro antitumoral activity is observed at millimolar concentrations [37]. Hence, a fundamental research inquiry is to determine the metformin concentrations achieved in tumors of patients receiving conventional antidiabetic metformin dose.

## 2. Mechanisms of Action

### 2.1. Complex I Inhibition

In the early 2000s, the inhibition of the mitochondrial respiratory chain Complex I emerged as a prominent mechanism explaining metformin’s anti-gluconeogenic effect in the liver of T2DM patients. Metformin binds to Complex I, inducing a mild and reversible non-competitive inhibition [26,38], contrary to typical inhibitors of Complex I, such as rotenone and piericidin A, which are uncharged, hydrophobic molecules that exhibit high efficiency (with an IC_50_ of approximately 2 μM), being able to halve Complex I activity at very low concentrations [39].

The precise mechanism by which metformin targets Complex I and exerts its function remains under investigation. In 2014, Bridges et al. [40] demonstrated that metformin interacts with the Cys39-containing matrix loop of the respiratory chain subunit ND3. They observed that metformin binds to Complex I in a deactive-like open-loop conformation, stabilizing the enzyme in an inactive state. Consequently, this inhibition leads to a lower proton gradient and reduced adenosine triphosphate (ATP) synthesis due to diminished nicotinamide adenine dinucleotide (NADH) oxidation, proton pumping across the inner mitochondrial membrane, and oxygen consumption rates. In 2023, the same group identified up to three potential binding sites for biguanides within various protein subunits of Complex I [41]. The primary inhibitory site is situated within the amphipathic region of the quinone binding channel (Q-channel), adjacent to a mobile structural element within the NDUFS7 subunit [18]. Mechanistically, these studies revealed that synthetic biguanide binding at this site prevents Complex I reactivation. Consequently, this inhibition leads to a moderate reduction in ATP synthesis in the liver, as NADH cannot transfer its electrons to Complex I. This limitation thereby reduces mitochondrial respiratory chain activity, leading to a decrease in the ATP-dependent gluconeogenic process (Figure 1).

Several mechanisms have been proposed to explain how Complex I inhibition decreases hepatic gluconeogenesis (Figure 1). These include alterations in the hepatic energy state, characterized by a moderate decrease in ATP/AMP and ATP/ADP ratios. These hepatic energy alterations underlie the inhibition of gluconeogenesis through AMPK-dependent and AMPK-independent pathways. Indeed, given that gluconeogenesis is a highly energy-consuming process, the hepatic energy charge alteration could be sufficient to reduce the gluconeogenic pathway [42,43,44].

### 2.2. Adenosine Monophosphate-Activated Protein Kinase (AMPK)-Dependent Mechanisms

AMPK activation may be one of the mechanisms explaining metformin’s effect on gluconeogenesis. It is suggested that alterations in hepatic energy status, like those resulting from Complex I inhibition, and increased AMP levels induce allosteric activation of AMPK [45,46], which leads to the phosphorylation and nuclear exclusion of the cAMP response element-binding protein (CREB)-regulated transcription co-activator 2 (CRTC2) with the subsequent gluconeogenic gene inhibition [47]. Notably, other mechanisms to inactivate CRTC2 by metformin have been described [34,48]. AMPK has also been shown to enhance the upregulation of the orphan nuclear receptor small heterodimer partner (SHP), which through direct or indirect interaction with CREB, suppresses CREB-dependent gene expression involved in hepatic gluconeogenesis [48].

AMPK activation by metformin remains a subject of controversy, with some studies suggesting its independence from energy status, as indicated by findings that activation relies on liver kinase B1 (LKB1) or calcium/calmodulin protein kinase 2 (CAMKbeta) [45]. Nonetheless, Foretz et al. (2019) [4] demonstrated that LKB1 knockout hepatocytes still respond to metformin’s effects. Furthermore, they demonstrated that the specific deletion of hepatic AMPK does not suffice to impede metformin’s action, suggesting the existence of AMPK-independent mechanisms facilitating metformin’s efficacy. Controversy surrounds these findings due to methodological variations, especially in in vitro studies with differing concentrations. Notably, only suprapharmacological levels of metformin inhibit Complex I in isolated mitochondria [49], making clinical extrapolation challenging. In addition, localization within hepatic cells is debated, with some studies suggesting mitochondrial accumulation and others cytosolic predominance [50]. The prevailing accepted hypothesis stands that metformin’s positive charge facilitates slow entry into mitochondria, driven by an electrochemical gradient, which leads to millimolar concentrations in the matrix. This gradual accumulation of metformin within the mitochondria contributes to its mild inhibitory effects on Complex I. This accumulation reduces membrane potential, self-limiting further uptake [15,26,40].

Metformin-induced AMPK activation may reduce gluconeogenesis by enhancing hepatic insulin sensitivity. This activation leads to the phosphorylation and inhibition of acetyl-coenzyme A (CoA) carboxylase 1 (ACC1) and acetyl-CoA carboxylase 2 (ACC2), resulting in decreased hepatic lipogenesis. Additionally, it promotes hepatic fatty acid oxidation by activating carnitine palmitoyltransferase 1 (CPT1), facilitating the transport of acyl-CoA into mitochondria [19,51]. As a result, metformin treatment would alleviate hepatic steatosis, thereby enhancing insulin sensitivity and, consequently, insulin’s ability to inhibit gluconeogenesis.

#### Adenosine Monophosphate-Activated Protein Kinase (AMPK)-Dependent Lysosomal Pathway

Low doses of metformin have been reported to activate AMPK through an AMP-independent mechanism involving the lysosomal protein complex v-ATPase-Ragulator in hepatocytes (Figure 2). This complex was described as an endosomal docking site for AXIN 1/LKB1 to mediate AMPK activation [52,53]. This mechanism relies on the interaction of presenilin enhancer 2 (PEN2) with the biguanide moiety of metformin; PEN2 is then recruited to the accessory protein v-ATPase ATP6P1 and v-ATPase is inhibited. As a result of v-ATPase inhibition, AXIN-LKB1 translocation to the lysosome surface is promoted, which leads to AMPK activation in the lysosome [54]. This novel mechanism may be particularly relevant in the intestine and liver, since the deletion of PEN2 abolishes the beneficial effects of glucagon-like peptide 1 (GLP1) secretion in mice enterocytes and blunts the reduction in hepatic lipid content in mice fed a high-fat diet [54].

This AMP-independent mechanism is of key importance, as it explains how low doses of metformin can activate AMPK, while AMP-dependent mechanisms require higher doses. As previously emphasized by Foretz et al. in a prior review, numerous unanswered questions remain to be addressed. This is particularly relevant, given that lysosomal activation of AMPK could potentially elucidate the effects of low doses of metformin in the intestine but not in the liver [18].

### 2.3. Adenosine Monophosphate-Activated Protein Kinase (AMPK)-Independent Mechanisms

Metformin also diminishes hepatic gluconeogenesis by directly inhibiting mitochondrial glycerol-3-phosphate dehydrogenase (mGPDH) in an energy- and AMPK-independent manner [17,55]. This enzyme, a component of the glycerol phosphate shuttle system, works in conjunction with its cytosolic isoform (cGPDH) to channel electrons from cytosolic NADH to the mitochondrial respiratory chain (Figure 3). Thus, the inhibition of mGPDH leads to the accumulation of NADH in the cytosol, which hinders gluconeogenesis from redox-dependent substrates such as glycerol and lactate, but not from redox-independent substrates like pyruvate, alanine, and dihydroxyacetone phosphate (DHAP). It is noteworthy that this selective inhibition of certain substrates’ utilization for gluconeogenesis may explain why metformin has a low risk of inducing hypoglycemia or does not alter gluconeogenesis in non-diabetic patients, where amino acid contribution to gluconeogenesis is higher. Nonetheless, some studies suggest that inhibiting the glycerol phosphate shuttle alone in the liver may not be sufficient to lower blood glucose levels. This discrepancy may result from the liver’s reliance on the malate–aspartate shuttle as the primary NADH transporter, which effectively reduces blood glucose levels when inhibited by metformin [56,57,58].

The mechanism by which metformin increases the NADH/NAD+ ratio remains debated. Conflicting findings suggest that this effect may stem from Complex I or mGPDH inhibition. Alternative proposals have emerged, including one positing that metformin accumulation in the mitochondrial matrix depolarizes the membrane, inhibiting the aspartate transporter of the aspartate–malate shuttle. Consequently, this inhibition may stimulate glycerol phosphate shuttle activity, reducing glycerol-3-phosphate (G3P) concentration, a potent allosteric inhibitor of phosphofructokinase 1 (PFK1), thus alleviating its inhibition and favoring glycolytic activity over gluconeogenesis [33].

Another AMPK-independent mechanism is based on the fact that the mild increase in intracellular AMP levels caused by metformin is sufficient to decrease hepatic gluconeogenesis stimulated by glucagon. AMP inhibits the glucagon-induced activation of adenylate cyclase, thereby reducing intracellular cyclic AMP levels. This action diminishes protein kinase A (PKA) activity and lowers the phosphorylation levels of key gluconeogenesis-related enzymes such as fructose-2,6-biphosphatase 1, the inositol trisphosphate receptor, and CREB1 [43,44]. Consequently, these events result in a reduction in glucagon-stimulated glucose production. Additionally, Miller et al. proposed that AMP allosterically inhibits fructose-1,6-biphosphatase, a key gluconeogenic enzyme [59]. Indeed, a recent study revealed that the expression of a mutant fructose 1,6-bisphosphatase enzyme, unresponsive to AMP regulation, nullified the glucose-lowering impact of metformin in vivo [43].

### 2.4. Complex IV Inhibition

Another proposed mechanism for the interaction of metformin with mitochondria suggests that metformin inhibits Complex IV, leading to the disruption of the OXPHOS system. This disruption alters cellular energetics, decreasing ATP production like Complex I inhibition, and indirectly reduces the ubiquinone pool, which serves as the electron acceptor for mGPDH, thus inhibiting mGPDH activity (Figure 3). The interaction between metformin and Complex IV may be facilitated by the ability of biguanides to bind metal ions, such as iron and copper, both of which are present in Complex IV [60]. However, it is worth noting that all complexes of the electron transport chain contain iron and/or copper ions, essential for electron transfer. Consequently, further research is needed to confirm whether metformin is capable of inhibiting the mitochondrial respiratory chain at different complexes beyond Complex I.

### 2.5. Epigenetic Effects of Metformin

Metformin has been shown to influence epigenetic modifications, particularly through its effects on DNA methylation and histone modifications. These metformin-induced epigenetic alterations underscore its potential as a promising therapeutic intervention with wide-ranging implications for disease management and prevention.

#### 2.5.1. Effects of Metformin on the Acetylation Profile

Metformin activates AMPK, which influences histone acetylation by regulating histone acetyl transferases (HAT). In particular, AMPK inhibits ACC, increasing intracellular acetyl-CoA levels available for histone acetylation [61]. This impacts gene expression, including gluconeogenesis genes like peroxisome proliferator-activated receptor gamma coactivator 1-alpha (PGC-1α) and phosphoenolpyruvate carboxykinase 1 (Pck1), potentially contributing to metformin’s antigluconeogenic actions [62]. Additionally, metformin may target the regulatory subunit of protein phosphatase 1 (PPP1R3C) to suppress cAMP-stimulated hepatic gluconeogenesis. Xueying Ji et al. propose that PPP1R3C overexpression could enhance proton pump interactor isoform 1 (PPI1) activity and modulate the phosphorylation of the CREB coactivator target of rapamycin complex-2 (TORC2) to upregulate the transcription of gluconeogenic genes [63]. Moreover, although there is lack of data, a study addressing the impact of histone acetylation in cancer research reported the ability of metformin to rectify specific histone H3 acetylation patterns in cancer-prone cells [63]. Nevertheless, further efforts need to be made to delve into the specific underlying mechanism and implications in clinical practice.

Metformin can also alter acetylation by affecting the activity of histone deacetylases (HDACs), which regulate gene expression by altering chromatin accessibility and modulate cellular processes via interaction with repressor complexes and transcription factors as well as deacetylation of non-histone proteins [64,65]. Sirtuins, a type of class III HDACs, are NAD+-dependent enzymes that mimic calorie restriction’s metabolic effects [66]. Metformin may enhance mitochondrial function partly through sirtuin-mediated pathways [67,68,69]. SIRT1, the most conserved mammalian class III HDAC [70], senses cellular energy levels and is directly activated by metformin. SIRT1 leads to AMPK-induced nicotinamide phosphoribosyltransferase (NAMPT) phosphorylation, with the subsequent NAD+ synthesis [71]. Furthermore, in silico and in vivo assays suggest that metformin could directly interact with the activation domain of SIRT1, thereby increasing SIRT1 catalytic efficiency under low NAD+ conditions [66]. SIRT1, in turn, activates AMPK through LKB1 Lys-48 deacetylation, forming a positive feedback loop [71]. Previous studies indicate that metformin can also exert its anti-hyperglycemic effects by modulating SIRT1. In detail, increased SIRT1 activity is linked to diminished gluconeogenesis through targeting TORC2 inhibition and enhanced gluconeogenesis via PGC-1α deacetylation [72]. Moreover, SIRT1 activity confers benefits on hepatic metabolism, mitochondrial biogenesis [67], inflammation [73], and diabetic nephropathy [74]. Conversely, in cancer cells, it has been implicated in potential contributions to mitochondrial dysfunction and pyroptosis [68].

HDAC Sirtuin 3 (SIRT3) is also a target of metformin. Primarily localized in the mitochondrial matrix, SIRT3 modulates mitochondrial dynamics, the tricarboxylic acid cycle, respiratory chain, and fatty acid oxidation [69]. Additionally, antioxidant effects of SIRT3 have been reported [75]. Metformin has demonstrated the ability to enhance mitochondrial function [76], beta-oxidation [75,77], and antioxidant defenses [76,78] by modulating SIRT3 activity. Additionally, metformin-mediated induction of SIRT3 may confer protection against chemotoxicity and exacerbate metformin-induced apoptosis and mitochondrial dysfunction in cancer cells [79].

#### 2.5.2. DNA Methylation

In addition to modulating histone acetylation/deacetylation, current data support a role for metformin as an epigenetic regulator by linking cellular metabolism to the DNA methylation machinery. Metformin increases the ratio of SAM/SAH by indirectly activating S-adenosylhomocysteine (SAH) hydrolase, thereby relieving the inhibition of S-adenosylmethionine (SAM)-dependent methyltransferases [80,81]. Genome-wide alterations in the DNA methylome of non-cancerous [80] and cancerous cells [81] in response to metformin treatment have been reported. In T2DM patients, three methylation sites (PBX/Knotted 1 Homeobox 2 (PKNOX2), WD and Tetratricopeptide Repeats 1 (WDTC1), and MHC Class I Polypeptide-Related Sequence B (MICB)) have been linked to the effects of metformin on HbA1c levels [82]. Additionally, higher DNA methylation levels in OCT1 (SLC22A1), an epigenetic change associated with hyperglycemia and obesity, are modulated by metformin [83].

### 2.6. Effects of Metformin on microRNAs 

Several microRNA (miRNA) circulating levels have been associated with diabetes and insulin resistance. Interestingly, an increase in Dicer 1, Ribonuclease III (DICER1) levels, an enzyme involved in miRNA maturation, has been observed in diabetic humans treated with metformin, as well as in mice. This observation suggests that metformin may exert its effects through the modulation of miRNAs [84]. Recent studies have identified circulating miRNAs altered by metformin treatment in T2DM patients, with miR-194-5p and miR-148-3a being of special interest due to their impact on Wingless-type mouse mammary tumor virus (MMTV) integration site family (Wnt) and Nuclear factor kappa-light-chain-enhancer of activated B cells (NF-kB) signaling pathways, both involved in T2DM development [85]. Additionally, in a cross-sectional study, metformin treatment was associated with changes in the circulating profiles of miR-192, miR-222, and mir-140-5p, paralleled by decreases in fasting glucose and HbA1c levels [86]. Circulating levels of miR-192 have previously been suggested as potential biomarkers of T2DM and miR-222 has been associated with the progression of obesity and insulin resistance [87]. Metformin also enhances hepatic redox balance by elevating the ratio of reduced glutathione to oxidized glutathione. Consequently, this alteration inhibits genes associated with gluconeogenesis through the let7–tet methylcytosine dioxygenase 3 (TET3)- hepatocyte nuclear factor (HNF)-4α pathway [88]. Moreover, in HepG2, miR-140-5p overexpression induced impairment of glucose consumption and glucose uptake [89].

### 2.7. Effects of Metformin on the Microbiota

Preclinical research has shown that metformin induces changes in the composition and function of the gut microbiota [90]. Evidence suggests potential benefits for metabolic and immune health resulting from these changes. However, the impact of metformin on the human gut microbiota remains ambiguous and additional research is required to clarify the mechanisms by which the drug operates and to comprehend its role in fostering the growth of particular bacteria that offer advantages to the host.

In a 2023 systematic review encompassing 13 studies, the use of metformin was linked to changes in bacterial genera abundance. However, data across different *phyla* were inconsistent, possibly due to differences in metformin dosage and treatment duration. Nonetheless, prior investigations found an enrichment in bacterial genera linked to beneficial effects on glucose metabolism and insulin sensitivity, such as *Lactobacillus* and *Akkermansia muciniphila* [91], following metformin treatment. An enrichment in *Akkermansia muciniphila* has been documented in both mice [92] and humans treated with metformin [93]. Similarly, a study on a T2DM mouse model revealed that metformin alone increases the abundance of *Lactobacillus* [94]. Moreover, in a randomized trial in overweight/obese adults, metformin treatment resulted in significant changes in the microbiota composition, increasing *Escherichia coli* and *Ruminococcus torques*, while decreasing *Intestinibacter bartlettii* and *Ruminococcus*. Remarkably, these changes in microbiota composition were associated with an increase in short chain fatty acids (SCFAs). The increased abundance of SCFA-producing bacteria has been postulated to facilitate SCFA-induced GLP1 secretion [95].

Several studies show a decrease in the abundance of *Bacteroides fragilis* [94,96,97]. *Bacteroides fragilis* produces the bile acid glycoursodeoxycholic acid (GUDCA), an endogenous antagonist of farnesoid X receptor (FXR), and it is hypothesized to be able to modulate GLP1 [97]. Notably, available data suggest that metformin can directly inhibit the growth of *Bacteroides fragilis* through the inhibition of the bacterial NADH–menaquinone oxidoreductase (NDH1) complex, a mechanism closely related to the eukaryotic mitochondrial respiratory chain Complex I, thereby interfering with ATP production [4,98].

### 2.8. Effects of Metformin as an Anticancer Agent

The antineoplastic effects of metformin have been mainly attributed to two mechanisms of action, the insulin-dependent and the insulin-independent mechanism. Within the insulin-independent framework, metformin exerts its effects predominantly through the inhibition of mammalian target of rapamycin (mTOR) activity. Specifically, the activation of AMPK phosphorylates tuberin 2 (TSC2), thereby stabilizing the TSC1-TSC2 complex and subsequently suppressing mTOR activation [99,100]. Furthermore, metformin’s modulation of fatty acid synthase (FAS) [101] and cyclin D1 [102] via AMPK activation has been implicated in inhibiting cell proliferation, further underlining its antineoplastic potential.

Conversely, the insulin-dependent effects of metformin are regarded as indirect, as they do not hinge on Complex I inhibition and AMPK activation. Here, the PI3K/Akt/mTOR pathway assumes paramount importance, as insulin-like growth factor 1 (IGF-1) and insulin govern cell survival and growth through this signaling cascade [103,104,105]. Metformin-mediated inhibition of hepatic gluconeogenesis and promotion of peripheral glucose absorption results in reduced insulin/IGF-1 levels and blood glucose [106]. Moreover, metformin downregulates the expression of insulin and IGF-1 receptors, leading to inhibition of AKT/mTOR/mitogen-activated protein kinases (MAPK)/extracellular-sgnal-regulated kinase (ERK) signaling [107,108]. Thus, the reduction in the insulin/IGF-1 signaling axis could explain the decrease in growth and mitogenesis.

Metformin has also been reported to exert anti-tumoral effects by promoting an effective immune response against tumor suppression. Cancer cells efficiently evade immune surveillance by modulating immune checkpoint molecules [109,110,111]. These molecules play a crucial role in balancing immune activity to prevent autoimmunity and minimize additional tissue damage. Cha et al. [112] demonstrated that metformin enhances cytotoxic T lymphocyte (CTL) activity by reducing the stability and membrane localization of programmed death-ligand 1 (PD-L1). Additionally, metformin activates AMPK, leading to direct phosphorylation of S195 on PD-L1. This phosphorylation induces abnormal PD-L1 glycosylation, causing its accumulation in the endoplasmic reticulum (ER) and subsequent degradation. Consistently, breast cancer tissues from patients treated with metformin show decreased PD-L1 levels alongside AMPK activation. Blocking PD-L1’s inhibitory signal with metformin boosts CTL activity against cancer cells through the ER-associated degradation (ERAD) pathway and suggests that combining metformin with CTLA4 blockade could enhance immunotherapy efficacy.

Additionally, the potential anti-cancer properties of metformin have been linked to the regulation of numerous miRNAs. A great effort has been made in the discovery of metformin-modulated miRNAs and the associated downstream effectors in various malignancies. Of note, miR-192 is a known prognostic biomarker in gastric cancer and miR-194 direct target Bmi-1 overexpression has been reported in different malignancies [113]. Thus, the anti-tumoral mechanisms of metformin could be mediated through miRNAs simultaneously involved in the anti-hyperglycemic actions of metformin.

Other protective mechanisms of metformin in cancer include epigenomic modifications. As discussed above, metformin can regulate the activity of several enzymes involved in epigenetic changes, such as HDACs [68,69], HATs [64], and the DNA methylation machinery [80,81].

## 3. Metformin in Cancer Risk and Treatment

An increasing number of epidemiological and clinical investigations have indicated that metformin decreases the risk of cancer in individuals with T2DM and enhances the survival outcomes of cancer patients diagnosed with breast, ovarian, liver, and colorectal tumors [22,24,25,114].

One proposed mechanism for the antineoplastic action of metformin involves the inhibition of mitochondrial respiratory Complex I. This inhibition compromises ATP production, consequently activating AMPK in an LKB1-dependent manner. This AMPK activation leads to mTOR inhibition, leading to anticancer effects, including reduced protein and lipid synthesis, slow proliferation rates, activation of autophagy, and inhibition of inflammatory responses.

According to public statistics from 2020, breast and colorectal cancers rank among the most prevalent types of cancer. Data from the Global Cancer Observatory (GLOBOCAN) of the International Agency for Research on Cancer (IARC) predict a significant increase in the global burden of breast and colorectal cancers by 2040, with over 3 million new cases per year and 1 million and 1.6 million deaths per year, respectively [115,116,117]. These findings emphasize the urgent need to advance new therapies and strategies to enhance cancer patient management. Repurposing metformin for cancer treatment holds promise for rapid progress in this regard, given its approved status, well-established pharmacokinetics, and tolerability profile. In fact, observational studies and systematic reviews indicate that metformin treatment in diabetic patients may lower cancer risk and mortality rates by 10% to 40%. Current research explores if similar benefits extend to non-diabetic cancer patients or those with impaired fasting glucose levels [118].

### 3.1. Breast Cancer

The estimated global incidence in 2020 of female breast cancer was around 2.3 million new cases, accounting for 11.6% of the total new cases, and emerged as the leading cause of cancer-related deaths, surpassing lung cancer [119].

The complex relationship between diabetes and breast cancer, both significant global health concerns, has stimulated extensive research to reduce risk and improve outcomes. Diabetes, a prevalent chronic disease, significantly increases the risk of breast cancer, which remains the leading malignancy among women worldwide [120]. Within this complex interplay, metformin, a widely used anti-diabetic drug, has become a focal point due to its potential anti-cancer properties.

#### 3.1.1. Clinical Studies

Epidemiological studies form the basis for understanding the relationship between diabetes, breast cancer, and metformin. Notable, women with T2DM are more likely to be diagnosed with breast cancer. This association persists even after adjustment for body mass index (BMI) and menopausal status [121].

Significantly, several observational studies have consistently shown a decreased risk of breast cancer and lower recurrence rates among individuals who are overweight and/or have T2DM and use metformin [122,123]. Metformin emerges as a significant player associated with improved disease-free survival (DFS) and overall survival (OS), suggesting its potential positive impact on survival outcomes among invasive breast cancer patients with T2DM [124]. In addition, a meta-analysis of 31,031 breast cancer patients suggests a potential benefit of metformin therapy, showing reductions in all-cause mortality and progression-free survival compared with non-metformin users [125].

However, while certain studies show beneficial effects of metformin on breast cancer risk, others report no association or nuanced effects compared with other cancers [126].

Lu et al. observed a higher likelihood of breast cancer diagnosis among women with T2DM but found no statistically significant reduction in breast cancer risk among metformin users [121].

Despite these nuanced findings, a consistent association between metformin exposure and reduced risk of hormone receptor-positive (HR+)/human epidermal growth factor receptor 2-negative (HER2) breast cancer is evident [123]. Another study exposes the significant limitations of real-world observational studies in accurately assessing metformin’s effects on cancer incidence and outcomes due to avoidable biases that, surprisingly, persist today [127]. These controversial studies underscore the importance of stratifying patients by molecular subtype to gain a clearer understanding of the effects of metformin within more homogeneous cohorts. Moreover, these conflicting results are similarly observed in studies investigating the potential of metformin as a chemopreventive or adjuvant agent [127,128].

Clinical trials are the main platform for evaluating the efficacy of metformin in breast cancer. These trials shed light on its impact on DFS and OS, particularly among invasive breast cancer patients with T2DM. These studies consistently show significant associations, suggesting that metformin is a potential game-changer in adjuvant therapy [123]. A systematic meta-analysis study of nine clinical trials and over 1000 patients provides nuanced insights, highlighting metformin’s capacity to decrease insulin levels, fasting blood sugar, BMI, and antigen kiel 67 (Ki-67) in patients with breast and endometrial cancer [129]. Thus, clinical analyses go beyond traditional survival metrics to examine plasma biomarkers, shedding light on metformin’s role in modulating hormonal and metabolic factors that affect breast cancer. Leptin, Homeostatic model assessment for insulin resistance (HOMA-IR), sex hormone-binding globulin (SHBG), and estradiol levels show positive responses to metformin, especially when combined with lifestyle interventions [130].

Clinical trials exploring metformin’s potential in combination therapies present a mixed picture. Some trials demonstrate safety and tolerability [131], even demonstrating that metformin in combination with chemotherapy (NCT01310231) improves the progression free survival and the clinical response of patients, ameliorating the adverse effects and enhancing the quality of life of breast cancer patients. However, other clinical trials, like a phase I clinical trial evaluating erlotinib and metformin, reveal no significant clinical benefits in metastatic triple-negative breast cancer (TNBC) patients [132]. In this sense, other clinical trials (NCT00930579 and NCT01101438) have revealed no significant effects in tumor proliferation or in the overall survival in breast cancer patients. Research also uncovers metformin’s therapeutic opportunity in controlling toxicities caused by neoadjuvant chemotherapy in non-diabetic breast cancer patients [133]. However, the intricacies of its impact on cholesterol levels after neoadjuvant treatment hint at broader metabolic effects [134].

#### 3.1.2. Animal Studies

Recent in vivo experiments conducted by Schmidt et al. provided dynamic insights into metformin’s effects on breast cancer progression. Combined treatment with a ketogenic diet and metformin in mouse models of TNBC shows significantly reduced tumor burden, increased tumor latency and slower tumor growth. This synergistic approach is emerging as a promising avenue for potentially prolonging survival [135].

In addition, TNBC xenografts show discernible changes in tumor imaging metrics following metformin treatment, highlighting its impact on tumor biology [136]. Furthermore, metformin has been shown to be synergistic with immunotherapy against TNBC in mouse models [137]. Studies conducted in metformin-treated mice underscore its immunomodulatory effects. Specifically, a reduction in M2-like macrophages, monocytic myeloid-derived suppressor cells (M-MDSCs), and regulatory T cells (Tregs) indicates the ability of metformin to enhance local antitumor activity [138]. Taken together, these studies suggest avenues for metformin’s potential in combination treatments. Finally, metformin’s cardioprotective effects are underscored in experiments aimed at attenuating doxorubicin-induced cardiotoxicity [139].

#### 3.1.3. In Vitro Studies

In vitro experiments are exploring the intricate cellular and molecular mechanisms underlying the effects of metformin on breast cancer cells. It is well known that altering mitochondrial function by inhibiting Complex I is an outstanding anticancer mechanism [140]. A remarkable consideration is that metformin antiproliferative effects may depend on glucose concentration in culture media [141]. It acts through mTOR-dependent and mTOR-independent pathways, demonstrating versatility in different cellular environments [141].

The metabolic reprogramming observed in cancer cells, particularly their ability to adapt metabolism to support rapid proliferation, is discussed in the light of metformin’s inhibition of mitochondrial Complex I. Mitochondrial pathways are identified as potential targets for therapeutic intervention, in line with the premise that understanding the metabolic context is essential [142,143].

The concentration- and time-dependent induction of apoptosis by metformin in breast cancer cells was demonstrated in a study by Gao et al., implicating the intrinsic mitochondria-mediated apoptosis pathway [144]. This highlights the potential of metformin as a cytotoxic agent against breast cancer. Furthermore, metformin’s influence on the antioxidant system and its modulation of various molecular factors, including pro- and anti-apoptotic proteins, matrix metalloproteinases-2 and 9 (MMP-2, MMP-9), miR-21, and miR-155, are highlighted by Sharma and Kumar [145]. These findings hold potential implications for targeted treatment and clinical management of breast cancer.

Metformin’s suppression of reactive oxygen species (ROS)/epidermal growth factor (EGF)-induced breast cancer cell migration, invasion, and epithelial–mesenchymal transition (EMT), through the inhibition of the PI3K/Akt/NF-κB pathway, highlights its potential as an anti-metastatic agent [146]. Studies are further elucidating its effects on breast cancer stem cells, highlighting its potential in targeting these initiating cells. Shi et al. highlighted the ability of metformin to suppress triple-negative breast cancer stem cells by targeting krüppel-like factor 5 (KLF5) for degradation, offering a promising avenue for treating aggressive forms of breast cancer [147].

In addition, the adjuvant potential of metformin in radiotherapy is emerging as a promising facet, as evidenced by its ability to increase radiosensitivity by modulating specific genes such as miR-21-5p and sestrin 1 (SESN1) [148].

Finally, cellular NAD+ depletion is emerging as a critical aspect of metformin sensitivity in breast cancer cells, as demonstrated in the study in Ref. [149]. NAMPT, a key player in maintaining NAD+ levels, influences metformin resistance and provides insights into potential targets for overcoming resistance in breast cancer treatment.

### 3.2. Colorectal Cancer

Colorectal cancer (CRC) ranks as the third most common cancer worldwide, with almost 2 million new cases diagnosed each year, accounting 9.6% of all cancer cases, and is the second leading cause of cancer-related deaths worldwide [120,121].

Obesity and T2DM are among the risk factors associated with CRC. Recent studies have proposed incorporating T2DM patients into CRC screening programs to mitigate their elevated risk [150,151]. Hyperinsulinemia and high levels of insulin-like growth factor 1 (IGF-1) are associated with increased proliferation of colon cells, resulting in malignancy [152]. The risk is even higher in patients treated with sulfonylureas and insulin [152].

#### 3.2.1. Clinical Studies

As previously mentioned, metformin is the first-line treatment for T2DM and emerges as a potential candidate for chemoprevention in cancer, including CRC. This is supported by epidemiological studies demonstrating that patients with T2DM not only have a reduced risk incidence of tumor development but also show a lowered risk of mortality when undergoing metformin treatment after CRC diagnosis [153,154,155]. More concretely, long-term consumption of metformin has been associated with a decreased risk of CRC in diabetic patients [156]. However, further studies are needed to demonstrate whether the observed chemopreventive effects in T2DM patients are due to the direct influence of metformin on cancer cells or due to the metabolic status provoked by the disease. In this sense, some studies have explored the effects of metformin in nondiabetic CRC patients with promising results [157]. Nonetheless, in other cancer types, such as prostate and breast cancer, the impact of metformin on nondiabetic patients appears to be non-significant [158,159,160].

Another interesting aspect to consider is the potential synergistic effect of metformin when combined with conventional chemotherapeutic agents or immunotherapy. In a recent study focusing on patients with stage IV microsatellite-stable CRC who had experienced progression on prior therapies (NCT03800602), Akce et al. demonstrated that the combination of nivolumab, a medication that binds to the programmed cell death protein 1 (PD-1) to enhance immune cell activity against cancer cells, along with metformin, did not exhibit significant immune modulation compared to metformin alone. However, the study revealed promising trends in tumorous T-cell infiltration following the dual treatment of metformin and PD-1 blockade, despite disease progression observed in the majority of patients [161].

Another study carried out in China by Zhang et al. analyzed a total of 187 T2DM patients who underwent resection of CRC [162]. The results unveiled inverse correlations between metformin treatment and the incidence of distant metastasis. as well as poorly differentiated adenocarcinoma grades. Furthermore, metformin inversely correlated with positive staining for a cluster of differentiation 133 (CD133) and β-catenin protein expression, a marker for cancer stem cells, and an upregulator of epithelial-to-mesenchymal transition, in patients with CRC and T2DM [162]. In addition to this, the combination of metformin with irinotecan resulted in disease control in 41% of patients within 12 weeks [163]. Moreover, the synergistic effects of metformin plus 5-fluorouracil (5-FU) have been reported in a phase 2 trial in patients with refractory CRC [164].

From another point of view, Kim et al. demonstrated that the glutamine metabolism could play an important role in the sensitivity of CRC patients-derived tumor–organoid cells to metformin [165]. It is known that glutamine metabolization by mitochondrial enzymes into alpha-ketoglutarate, promotes cell growth by the induction of mammalian target of rapamycin complex 1 (mTORC1) activation, in contrast to metformin effects [166]. Thus, a combination of glutamine metabolism inhibitors with metformin could be an effective adjunctive treatment against CRC. In another study, metformin modulated the mitochondrial metabolism in CRC patients-derived organoids, with effects in mitochondria morphology, with deformation and partial loss of mitochondrial membranes and degradation of cristae [167]. Consequently, the authors also observed an accumulation of free radicals and a loss of membrane potential, suggesting that metformin is a good candidate to act synergistically with other drugs to induce cell death [167].

It is imperative to consider a recent review that underscores the controversy surrounding the impact of metformin treatment on tumor cells in human patients. This review posits that the concentrations of metformin required for substantial inhibition of Complex I are roughly 1000 times higher than those typically utilized for treating T2DM, making its application as a neoadjuvant anticancer therapy impractical [37]. Indeed, clinical trials involving non-diabetic individuals highlight the need to refine dosing strategies. For instance, a prospective Japanese phase III trial discussed by Cunha Júnior AD et al. explored the potential of low-dose metformin over one year in non-diabetic individuals at high risk for new polyps, showing a reduction in polyp formation and colorectal adenomas [168]. This underscores the importance of redefining accurate dosing and patient monitoring to ensure the safe and effective use of metformin in cancer therapy. Despite the theoretical possibility of metformin accumulation in mitochondria owing to its cationic charge, there exists no definitive supportive evidence for this assertion. Conversely, metformin may exert effects through mechanisms distinct from Complex I inhibition. Studies have indeed suggested that while metformin diminishes the adenosine diphosphate/adenosine triphosphate (ADP/ATP) ratio and activates AMPK, such effects alone might not be sufficient to provoke significant energy depletion and cellular demise [49,169].

Taking together these considerations in human studies on metformin effects on CRC chemoprevention and as a possible combined therapy with classical chemotherapeutic agents or in immunotherapy, it must be considered the basic in vitro and in vivo studies to elucidate the molecular mechanisms of metformin, with a special focus on mitochondria.

#### 3.2.2. Animal Studies

Several animal studies support the potential anticancer effect of metformin against various types of cancer, including CRC. In murine models of CRC induced by chemical carcinogens or mutations in the adenomatous polyposis coli (APC) gene, administration of metformin-activated AMPK and inhibited the mTOR/ribosomal protein S6 kinase (S6K) pathway. This led to a decrease in colonic mucosal epithelial cell proliferation and a reduction in the development of intestinal polyps [170,171]. These results suggest that metformin holds promise as a potential candidate for CRC chemoprevention.

In addition to the chemopreventive properties, other in vivo findings suggest that metformin combined with adjuvant chemotherapy might be associated with a better prognosis. In a chemical-induced CRC model using 1,2-dimethylhydrazine (DMH), the combination of metformin and oxaliplatin decreased DMH-induced CRC in diabetic and non-diabetic mice by downregulating tumor angiogenesis [172]. Similarly, in patient-derived xenograft models, the administration of metformin in combination with 5-FU or 5-FU plus oxaliplatin exhibited a tumor-suppressive effect by activating AMPK-mediated pathways, which was accompanied by a reduction in stem-like cells [173,174]. A recent preclinical study revealed that metformin suppressed liver metastasis of a CRC cancer xenograft mouse model, mediated by the inhibition of mTOR phosphorylation through activation of AMPK [175].

It is noteworthy that the endocrine and metabolic alterations accompanying obesity are a factor to consider in promoting CRC, and the ability of metformin to correct some of these alterations may contribute to its antineoplastic effect. For instance, obese individuals exhibit reduced levels of adiponectin, which may contribute to the negative impact of obesity on neoplastic diseases. This adipokine activates AMPK, thereby inhibiting cell growth [176]. Therefore, metformin could be particularly beneficial in mitigating the adverse effects of obesity on neoplasia [177]. In fact, Algire et al. demonstrated in vivo that metformin inhibits the stimulatory effect of a high-energy diet on colon carcinoma growth [178]. The authors observed that metformin treatment reduced insulin levels, attenuated diet-induced phosphorylation of protein kinase B (AKT), decreased the expression of fatty acid synthase (FASN), and activated AMPK. These findings suggest that metformin may have a potential in addressing colon cancer in the context of obesity.

One interesting study conducted by Scharping et al. unveiled the role of metformin in altering the metabolism of the tumor microenvironment, thereby enhancing the response to PD-1 blockade immunotherapy in mouse tumor models [179]. Additionally, some studies have indicated that metformin can modulate the metabolic profile of gut microbiota [97,180]. Broadfield et al. (2022) showed that metformin reduces tumor growth in mice fed a high-fat diet, through changes in the gut microbiome, using a murine model of CRC and fecal transfer approaches [180]. In fact, the authors demonstrated the transfer of the gut microbiome, from metformin-treated mice to drug-naïve, conventionalized fed mice on a high-fat diet, results in the suppressed growth of murine colon cancer cells. Increased levels of SCFAs, such as butyrate and propionate, were found, which may be associated with down-regulation of highly activated T cell clusters (e.g., CD8+, NK1.1+, and Ki67+). Recent evidence also suggests the impact of butyrate in promoting CD8+ T cell long-term survival as memory cells [181].

#### 3.2.3. In Vitro Studies

Numerous in vitro studies have explored the efficacy of metformin and its underlying molecular mechanism in colon cancer cells. Initial evidence was provided by Zakikhani et al., showing that metformin reduces the proliferation of HT-29 cells in a dose- and time-dependent manner, by activating AMPK, a major regulator of the cellular energy metabolism [177]. Another study found that metformin transiently inhibits CRC cell proliferation by inducing G0/G1 phase arrest. The anti-proliferative effect may involve AMPK activation or increased ROS production [182]. Metformin inhibits Complex I activity of the mitochondrial electron transport chain, resulting in the mitochondrial depolarization and the release of ROS, which contribute to metformin antiproliferative effect.

Moreover, evidence suggests that metformin improves tumor sensitivity to chemotherapy drugs like 5-FU, oxaliplatin, or irinotecan, underscoring the involvement of mitochondria in the sensitization induced by metformin. Boyle and collaborators suggested that the activation of antitumor mitophagy by metformin could modulate the chemotherapy response in CRC cells [183]. This is also consistent with another study performed by Denise et al., showing that colon cancer cells reprogrammed their metabolism to oxidative phosphorylation (OXPHOS) in response to 5-FU and, only under these conditions (with 5-FU treatment), were colon cancer cells sensitive to metformin treatment [184]. These authors also determined that OXPHOS inhibition was the primary mechanism of action of metformin resulting in anti-tumor effects, while targeting AMPK played a marginal role. Zhang and collaborators found that metformin enhanced the sensitivity of CRC cells to cisplatin by modulating mitochondrial function, resulting in decreased membrane potential and induction of ROS production. This ultimately promoted apoptosis through the phosphoinositide 3-kinase (PI3K)/Akt pathway [185], providing an alternative mechanism to AMPK activation. Another study demonstrated that metformin-induced tumor suppression was prevented in cancer cells expressing NADH-ubiquinone reductase (H(+)-translocating), also known as NDI1, which is metformin-resistant yeast analogue of complex I. This suggests the crucial role of inhibiting this mitochondrial target in the drug’s anticancer effect [186]. This is also consistent with another work showing that metformin in CRC cells altered mitochondrial activity by increased ROS levels and sirtuin 3 (SIRT3) activity, suggesting that these changes may be crucial for its cytotoxic effects [187]. However, other studies suggest that metformin may mitigate DNA damage and mutagenesis by reducing ROS production [178,188,189].

Metformin, in combination with 5-FU, a first-line drug in CRC treatment, significantly enhanced the antiproliferative effect, apoptosis, and cell-cycle arrestment in SW620 cells, likely due to the activation of liver kinase B1 (LKB1)/AMPK pathway, leading to mTOR activity inhibition [162]. This combination may selectively target cancer stem cells (CSC) through the inhibition of the β-catenin pathway [190] or by AMPK activation and the inhibition of DNA replication, and the nuclear factor kappa-light-chain-enhancer of activated B cells (NF-κB) pathway [191]. CSC contributes to therapy resistance, thus, strategies targeting the depletion of this subpopulation of cancer cells are likely to be more effective in eradicating the tumor. In fact, synergistic effects of metformin plus 5-FU have been reported in a phase 2 trial [164]. Similarly, metformin combined with 5-FU and oxaliplatin restored sensitivity in chemoresistant CRC cells by reducing stemness and EMT through inhibition of the Wnt/β-catenin pathway [192].

As mentioned above and as shown in Table 1, metformin has been used as a single agent or in combination with other treatments both in vitro and in animal models, and the experimental data support numerous antidiabetic and anticancer properties.

## 4. Future Perspectives and Improvements in Cancer Therapy

The evaluation of metformin’s potential in cancer therapy has been marked by a shift in focus from randomized efficacy trials, which have tempered enthusiasm for its clinical use, to subsequent preclinical and clinical data revealing new avenues of research. While late-stage efficacy trials aiming to repurpose metformin as an anticancer agent have yielded discouraging results, recent well-designed clinical trials in specific patient populations have underscored the significance of patient stratification [193]. These trials suggest potential benefits when metformin is administered to selected groups, prompting exploration into its combinations with other therapies and its role as an early phase cancer prevention agent.

Nevertheless, uncertainties persist regarding the optimal cellular concentrations of metformin. This uncertainty stems from its low concentration and short half-life in plasma, coupled with its observed accumulation in tissues, including tumors [18]. These factors could potentially explain the discrepancy with the high concentrations utilized in preclinical studies. Understanding the differential response of tumor cells compared to normal cells to metformin is crucial, as is elucidating the mechanisms underlying tumor cell resistance to its effects. While metformin has demonstrated cytotoxic effects in various breast cancer cell lines, resistance has been observed in non-transformed breast epithelial cells MCF10A [194].

Remarkable, the translation of preclinical findings to clinical application faces challenges, particularly regarding metformin’s dosage and potential adverse events. High concentrations used in preclinical studies may exacerbate adverse effects associated with therapeutic use, including gastrointestinal upset, vitamin B12 deficiency, and hemolytic anemia [195]. Although metformin toxicity leading to hyperlactatemia and metabolic acidosis is rare, caution is warranted, especially in cases of overdose, as well as renal insufficiency [195]. Therefore, careful consideration of dosage and vigilant monitoring of patients are essential to mitigate these risks and facilitate the safe and effective use of metformin in cancer therapy. To address this challenge, researchers are exploring various metformin-based compounds to enhance its antitumor properties across different types of cancer.

Ongoing research is investigating novel, more lipophilic derivatives of metformin to target mitochondria. Gang Cheng et al. have developed metformin derivates called Mito-MET by attaching positively charged lipophilic substituents. These mitochondria-targeted analogues of metformin enhance antiproliferative and radio-sensitizing effects in cancer cells [196].

Similarly, other researchers have developed metformin derivates obtained by combination with other antitumor compounds studied, obtaining promising results. A combination strategy involving WZB117 (2-fluoro-6-(m-hydroxybenzoyloxy) phenyl m-hydroxybenzoate, a glucose transport protein 1 (GLUT1) inhibitor), OCMC (O-carboxymethyl-chitosan), and metformin was proposed for breast cancer treatment. WZB117–OCMC–metformin exhibited improved efficacy by simultaneously targeting GLUT1 and mTOR, overcoming limitations of metformin monotherapy [197]. Moreover, polyethylene glycol niosomal nanoparticles co-loaded with metformin and the widely used phytochemical artemisinin have shown enhanced anticancer effects on A549 lung cancer cells [198]. Similar results were observed by Kumar et al. by combining metformin and gallic acid, a phenolic acid found in tea leaves and some fruits with anticancer properties [199]. A novel compound, Met-ITC, was developed by incorporating an isothiocyanate moiety to metformin. The isothiocyanate (ITC), as a hydrogen sulfide (H2S) donor, acts as an anticancer agent by affecting the cell cycle, inducing apoptosis and inhibiting histone deacetylases. This hybrid molecule demonstrated enhanced efficacy and potency against various cancer cells (AsPC-1, MIA PaCa-2, and MCF-7) while being less effective on non-tumorigenic cells (MCF 10-A) [200].

Modifications to the metformin molecule could also provide novel drugs for the treatment of cancer as well as other diseases. For instance, supformin (LCC-12), a rationally designed dimer of metformin, exhibited anti-inflammatory effects by targeting the cell surface glycoprotein CD44 and regulating copper (II) levels in mitochondria. This intervention led to reduced NAD(H) redox cycling, metabolic and epigenetic reprogramming, and decreased inflammation in macrophages. LCC-12 demonstrated potential therapeutic effects in mouse models of bacterial and viral infections, highlighting its role in modulating cell plasticity and controlling inflammatory responses deregulated in cancer [201].

Due to its poor oral absorption, therapeutic doses of metformin are relatively high, often causing unpleasant gastrointestinal adverse effects. Consequently, novel derivatives of metformin have been synthesized over the past decades to overcome this limitation or achieve a more targeted release. Specifically, metformin-loaded liposomes have been designed for targeted delivery to inflamed endothelia. Using a three-step pretargeting system based on biotin–avidin interaction, these liposomes effectively delivered metformin hydrochloride to inflamed endothelial cells, offering potential anti-inflammatory therapy for conditions such as neurodegenerative diseases, atherosclerosis, and cancer [202].

Moreover, Zhao et al. are studying Met-3BP-Lip@M1, a drug delivery system derived from M1 macrophage membranes, which targets breast cancer cells to kill them by simultaneously inhibiting glycolysis and oxygen consumption. This biomimetic nanomedicine induced cancer cell apoptosis through effective cellular uptake and demonstrated synergistic improvements in therapeutic efficiency against breast cancer, both in vitro and in vivo [203]. In the same spirit, sulfenamides and sulfonamides metformin derivatives have displayed their potential as prodrugs and inhibitors of various diseases, including cancer [204].

Additionally, to improve the antitumor effects of metformin while reducing dosage requirements, a novel polymer dot, MA-dots, has been developed. These dots, synthesized using metformin and L-arginine, exhibit dual targeting capabilities for tumor cell membranes and mitochondria, demonstrating a 12-fold increase in antitumor activity compared to raw metformin, with effective tumor growth suppression in vivo [205]. Similarly, combining Mito-MET and iron chelators, such as deferasirox (DFX) and dexrazoxane (DXR), Cheng et al. showed synergistic inhibition of pancreatic and triple-negative breast cancer cell proliferation [206].

Addressing challenges in tumor immunotherapy, FCM@4RM, a tumor-specific nanovaccine, was developed to deliver tumor antigens and adjuvants while modulating the immune microenvironment. FCM@4RM demonstrates effective antigen presentation, stimulation of effector T cells, and modulation of the immune microenvironment through the synergistic effects of cytosine-phosphate-guanine (CpG), metformin, and a bioreconstituted cytomembrane, offering a novel approach for antitumor immunotherapy [207].

## 5. Concluding Remarks

In summary (Figure 4), the accuracy of drug levels achieved in cancer cells in laboratory data represents a challenge for applying findings to the clinical setting. Therefore, further clarification about mechanisms of action of metformin and its biodistribution to cancer cells is essential for designing novel metformin-based compounds, ultimately optimizing clinical applications and outcomes. Finally, combined therapies and targeted delivery strategies hold promise for improving the efficacy of and minimizing the adverse effects of metformin in cancer treatment.

## Figures and Tables

**Figure 1 biology-13-00302-f001:**
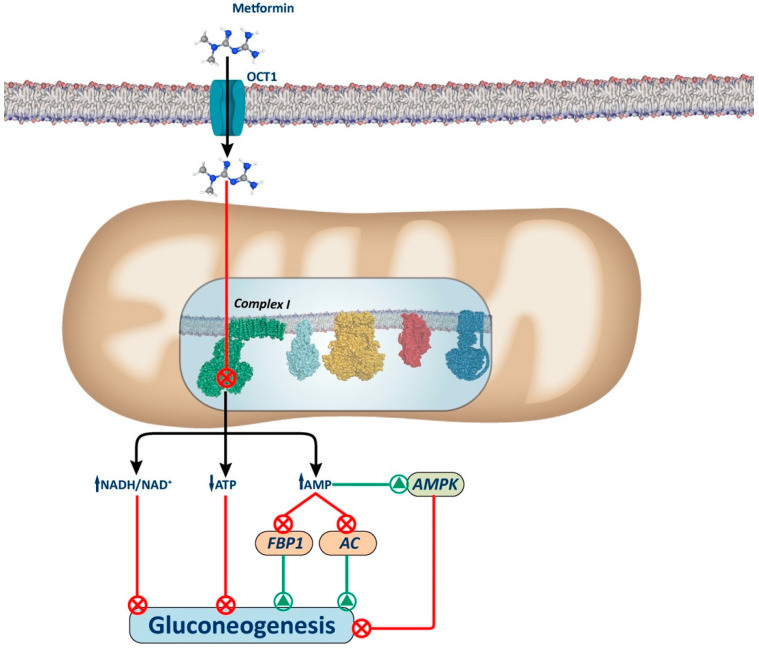
Complex I-dependent mechanism. The figure illustrates the proposed mechanism for metformin via inhibition of Complex I. Metformin inhibits complex I, causing a slight decrease in mitochondrial respiratory chain activity, resulting in an increase in the cellular redox state (NADH/NAD+) and a decrease in ATP synthesis (ATP/adenosine diphosphate (ADP)), which leads to the inhibition of gluconeogenesis. Additionally, there is an increase in AMP levels that induces gluconeogenesis inhibition by AMPK-dependent and AMPK-independent mechanisms. Among the AMPK-independent mechanisms, increased AMP levels can inhibit the activity of gluconeogenic enzymes, such as AC and FBP1. AC: adenylate cyclase; FBP1: fructose-1,6-biphosphatase.

**Figure 2 biology-13-00302-f002:**
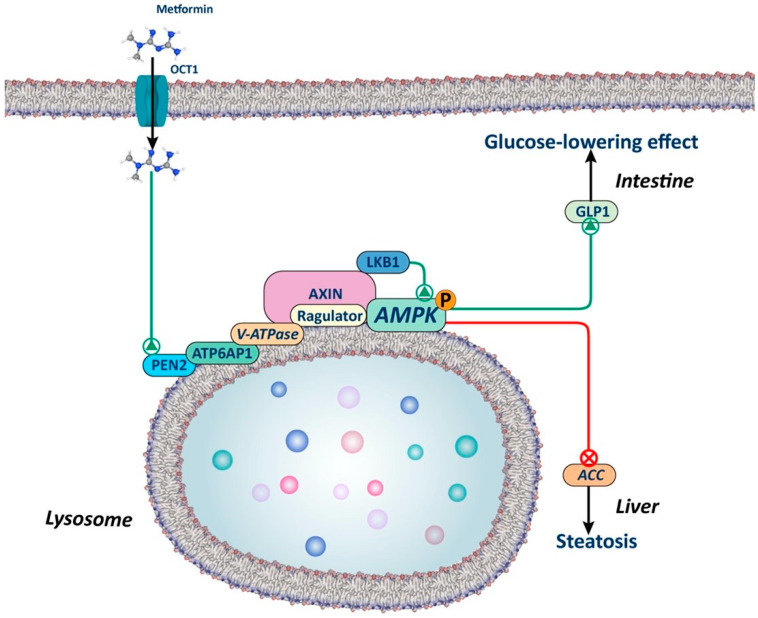
Lysosomal AMPK-dependent mechanism. The figure illustrates how metformin, through a lysosome-dependent mechanism, can activate AMPK. Metformin interacts with presenilin enhancer 2 (PEN2), which in turn is recruited by the v-ATP6P1 protein of the v-ATPase, leading to its inhibition. This inhibition promotes the translocation of AXIN-LKB1 to the surface of the lysosome, creating a docking site for AMPK activation. Studies in animals with silenced PEN2 suggest that this mechanism may play a significant role in the liver and intestine, regulating lipid metabolism and the secretion of glucagon-like peptide 1 (GLP1), respectively.

**Figure 3 biology-13-00302-f003:**
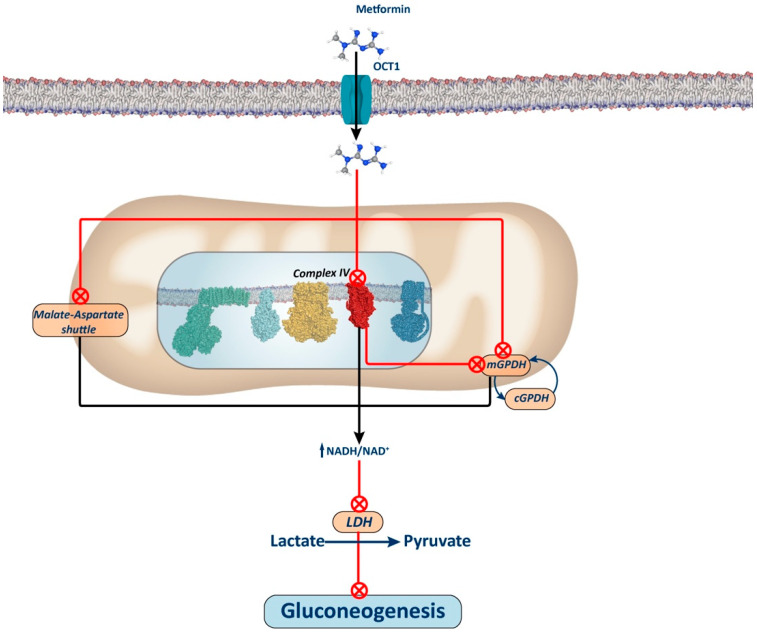
Complex IV-dependent mechanism and mitochondrial GPDH-dependent mechanism. The figure illustrates the proposed mechanism for metformin via inhibition of Complex IV. Inhibition of this complex leads to a reduction in the ubiquinone pool, subsequently inhibiting mGPDH activity and, thus, electron entry through the glycerol phosphate shuttle. It has been proposed that metformin may also directly inhibit this shuttle, thus contributing to an increase in the NADH/NAD+ ratio and thereby restricting the use of redox-dependent substrates, such as lactate and glycerol, for gluconeogenesis. Alternatively, it has also been proposed that the elevation of the NADH/NAD+ ratio induced by metformin through the inhibition of either Complex I or IV of the oxidative phosphorylation (OXPHOS) system may inhibit the aspartate–malate shuttle. In this scenario, inhibition of this shuttle would lead to compensatory activation of the glycerol phosphate shuttle, thereby decreasing glycerol-3P levels and releasing the repressive effect of this metabolite on phosphofructokinase 1. Consequently, this would enhance glycolysis, as opposed to gluconeogenesis (not depicted in the figure).

**Figure 4 biology-13-00302-f004:**
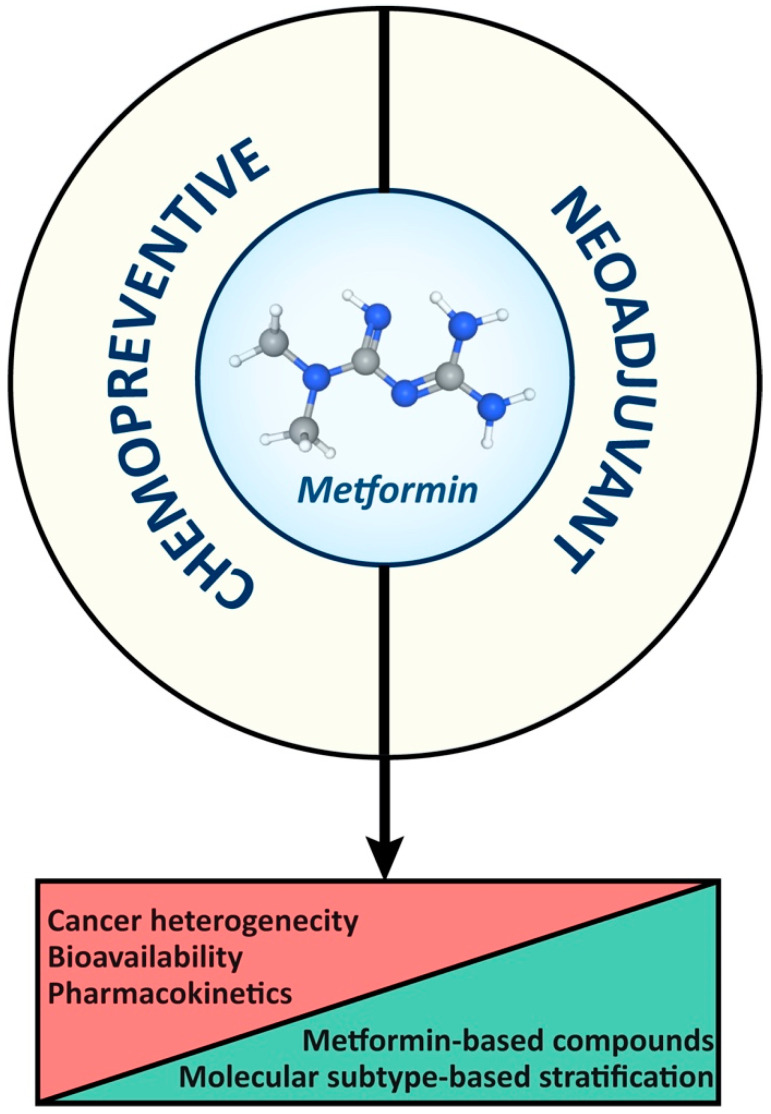
Constraints and opportunities of metformin as a chemopreventive and therapeutic agent in breast and colorectal cancer management.

**Table 1 biology-13-00302-t001:** Function of metformin in diabetes, breast cancer, and colorectal cancer.

Function/Results	Disease	In Vitro/In Vivo	References
Complex I inhibition	Cancer, Diabetes	In vitro	[26,38,40,41,140,142,143,182,186]
Apoptosis induction	Cancer	In vitro	[144,145]
Induction of MMP-2, MMP-9, miR21, and miR-155	Cancer	In vitro	[145]
PI3K/Akt/NF-kB pathway inhibition	Cancer	In vitro	[146,191]
Targeting KLF5 for degradation	Cancer	In vitro	
Modulates miR-21-5p and SESN1	Cancer	In vitro	[148]
Cellular NAD+ depletion	Cancer	In vitro	[149]
Reduction in cell proliferation by AMPK activation	Cancer	In vitro	[177,182]
Reduction in cell proliferation by G0/G1 cell cycle arrest	Cancer	In vitro	[162,182]
Activated ULK1 to stimulate an anti-tumor mitophagy program	Cancer	In vitro	[83]
Enhanced cisplatin-induced apoptosis dependent on the generation of ROS	Cancer	In vitro	[185]
Mitigation of DNA damage and mutagenesis by reducing ROS production	Cancer	In vitro	[178,188,189]
Complex IV inhibition	Diabetes	In vitro	[60]
Metformin-induced alterations in aspartate-malate shuttle and allosteric effectors of PFK1 and FBP1	Diabetes	In vitro	[33]
Activation of AMPK-dependent lysosomal pathway (AXIN/LKB1) at low doses: loss of GLP1 secretion upon PEN2 deletion	Diabetes	In vitro, in vivo	[53,54,185]
Enhancement of mitochondrial function and anti-hyperglycemic effects by SIRT1 and SIRT3 modulation	Cancer, Diabetes	In vitro, in vivo	[67,76,187]
Tumor burden, increased tumor latency, and slower tumor growth	Cancer	In vivo	[135]
Local antitumor activity by reduction in M2-like macrophages, M-MDSCs, and Tregs	Cancer	In vivo	[138]
Attenuation of doxorubicin-induced cardiotoxicity	Cancer	In vivo	[139]
Suppression of colorectal aberrant crypt foci and intestinal polyps development, as well as liver metastasis by activating AMPK	Cancer	In vivo	[170,171,175]
Metabolic changes decreasing oxygen consumption, activating AMPK pathway, and causing a reduction in cell growth	Cancer	In vivo	[173]
Downregulation of tumor angiogenesis and cell proliferation	Cancer	In vivo	[172]
Reduction in stem-like cell subpopulation	Cancer	In vivo	[174,190]
Inhibition of the stimulatory effect of a high-energy diet on tumor growth: reduced insulin levels and AKT and FASN expression; changes in the gut microbiome	Cancer	In vivo	[178,180]
Reduction in Ras-induced ROS production and associated DNA damage	Cancer	In vivo	[178]
Enhanced the effectiveness of anti-PD-1 immunotherapy by mitigation of tumor hypoxia	Cancer	In vivo	[179]
Reduced gluconeogenesis and hepatic lipid content by AMPK activation at suprapharmacological doses	Diabetes	In vivo	[47,48,51]
Impairment in glucagon-stimulated glucose production by reduction in PKA activity	Diabetes	In vivo	[43,44,59]
Suppression of cAMP-stimulated hepatic gluconeogenesis through HAT activity	Diabetes	In vivo	[63]
Reduced gluconeogenesis and HbA1c associated with specific miRNAs modulation	Diabetes	In vivo	[86,88]
Increase in SCFAs-producing bacteria	Diabetes	In vivo	[92,93,94,95]

## Data Availability

No new data were created or analyzed in this study. Data sharing is not applicable to this article.

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
