# Peer review of "Metformin: From Diabetes to Cancer—Unveiling Molecular Mechanisms and Therapeutic Strategies"

_biology, 2024, doi:10.3390/biology13050302_

Round 1

Reviewer 1 Report

Comments and Suggestions for Authors

In this review, the authors summarized the mechanism, utilities in diabetes and cancers of metformin in detail. The English are good and the review are in good structure. Below are a number of issues that the authors shall address or revise:

1. It is better to shorten the mechanism part and add more studies on the function of metformin in cancer.

2. Authors should make a table to summarize the function of metformin in different diseases and cancer types.

Reviewer 2 Report

Comments and Suggestions for Authors

 The manuscript titled "Metformin: From Diabetes to Cancer - Unveiling Molecular Mechanisms and Therapeutic Strategies", prepared by Amengual-Cladera E. et al., presents a review article covering literature over recent years (search depth 1991-2023), including results from multicenter clinical-epidemiological and experimental studies on exploring the therapeutic potential of metformin not only as an antihyperglycemic drug but also in combined cancer therapy. The work thoroughly discusses the pharmacokinetics of metformin and its various mechanisms of action on the body's energy balance by regulating gluconeogenesis in the liver pathways, implemented by complex I of the mitochondrial respiratory chain and AMPK. Sections dedicated to the role of metformin in the regulation of histone acetylation/deacetylation processes and DNA methylation, as well as its impact on microRNAs determining the drug's anticancer properties, are well presented. Speculations and evidence of metformin's role in regulating the number of normal intestinal microflora, which is associated with beneficial effects on glucose metabolism and insulin resistance, also significant in the effectiveness of cancer therapy, are expressed. A significant part of the review occupies data from a large array of studies on the anticancer effects of the drug and mechanisms of its action, crucial for the application of targeted treatment and clinical management of breast cancer and colorectal cancer. A separate section is dedicated to further prospects of using the antitumor properties of metformin and its compounds in cancer treatment.

The work is well illustrated, contains figures, and may represent a great scientific and practical interest. At the end of the manuscript, a reasoned conclusion is provided.

At the same time, the manuscript has a number of significant comments that require additions:

1. Despite the diverse approach of the authors of the manuscript to revealing the mechanisms of action and therapeutic potential of metformin in cancer patients at the present stage, there is no clear justification for its role and the possibility of use in specific groups of patients. The mechanisms of action of metformin and, accordingly, the stratification of treatment with selection of the dose of the drug in cancer patients without diabetes and in patients with type 2 diabetes mellitus when selecting therapy require clarification. The manuscript lacks comparisons of the effects of metformin and its dose on morbidity and mortality among patients with cancer, including breast and colorectal cancer. It is necessary to add links to sources that include this kind of clinical studies, which will enhance the information content and evidence of this study. Example: Chen K. et al. (2020) doi: 10.18632/aging.102787; Cunha Júnior AD et al. (2021) doi: 10.3748/wjg.v27.i17.1883.

2. The Conclusion section also requires rethinking and additions related to the use of metformin and compounds based on it as a combination therapy for breast and colorectal cancer, clarifying specific groups of patients with or without type 2 diabetes mellitus.

Reviewer 3 Report

Comments and Suggestions for Authors

This review focuses on a commonly used anti-diabetic drug, Metformin and how it has potential in managing other diseased state other than diabetes. This review highlights the current application of Metformin in different research area such as Epigenetics, microRNAs, and Microbiota and how it can be potentially applied in cancer research as well. Metformin has been studied for management of breast and colorectal cancer, with possibility of its role in management of other types of cancer as well. There has been debate on the underlying molecular mechanism of Metformin and this review highlights the need for unveiling the molecular mechanism by which, Metformin exerts its action. The review also highlights the ongoing research with Metformin with different cancer types and how different modified version of Metformin is being used to study its effect on various cancer types. Even with the current advances with Metformin in various cancer cell types, the study on molecular mechanism is still lacking and needs to be addressed. The authors have done a good job of highlighting the potential of Metformin in the field of cancer biology. However, I have some minor questions and suggestions regarding the review.

The authors have not provided definitions of all the acronyms used. For example, FXR and GDF15. The authors can provide list of all the abbreviations used.

There is another review on metformin and the authors of that review have slightly different take on the application of metformin. I was wondering if the authors of this review referred to that and have any thoughts about that. Below is the review article.

Lord, Simon R., and Adrian L. Harris. "Is it still worth pursuing the repurposing of metformin as a cancer therapeutic?." British Journal of Cancer 128.6 (2023): 958-966.

Are there more study on mechanism of action of Metformin through glycosylation? There are some articles that suggest that it may have effect on glycosylation as well. I was wondering if authors of this review have any comments on that as well. Below is the review article.

Cha, Jong-Ho, et al. "Metformin promotes antitumor immunity via endoplasmic-reticulum-associated degradation of PD-L1." Molecular cell 71.4 (2018): 606-620.

Reviewer 4 Report

Comments and Suggestions for Authors

The authors carried a review work of metformin, a traditional medicine for clinical diabetes, on its roles in cancer treatments. The manuscript added emphasis on the molecular mechanism of metformin effect, including inhibition of Complex I and IV in mitochondria, AMPK-dependent and independent mechanisms, and regulations at epigenetic, microRNA and microbiota levels. They further provided discussions on anticancer effect of metformin on breast and colorectal cancers by going through clinical, animal and in vitro studies. This review work can be helpful for researchers and clinical doctors.

Here are two points for improvement consideration:

1) The authors discussed multiple aspects of the working mechanism by metformin, however, these different mechanisms may be connected or subsequential to each other. It is suggested to add the internal connections among them, instead as isolated ones. 

2) Metformin shows cytotoxic effect on cancer cells, which can be imaginable on normal cells as well, particularly at higher using dosage than that for anti-diabetes. This side effect on health should be mentioned. 

Comments on the Quality of English Language

The English writing is well done in the manuscript, except for some minor ones like the fonts, and duplicated words. 
